# Fucoidan Isolated from *Saccharina japonica* Inhibits LPS-Induced Inflammation in Macrophages via Blocking NF-κB, MAPK and JAK-STAT Pathways

**DOI:** 10.3390/md18060328

**Published:** 2020-06-24

**Authors:** Jing Ye, Donghui Chen, Zhicheng Ye, Yayan Huang, Na Zhang, Edmund M. K. Lui, Changhu Xue, Meitian Xiao

**Affiliations:** 1College of Chemical Engineering, Huaqiao University, Xiamen 361021, China; cdh708@126.com (D.C.); yezc0122@163.com (Z.Y.); yyhuang@hqu.edu.cn (Y.H.); zhangna@hqu.edu.cn (N.Z.); mtxiao@hqu.edu.cn (M.X.); 2Xiamen Engineering and Technological Research Center for Comprehensive Utilization of Marine Biological Resources, Xiamen 361021, China; 3Physiology and Pharmacology, Western University, London, ON N6A 5B9, Canada; emk.lui@gmail.com; 4College of Food Science and Engineering, Ocean University of China, Qingdao 266003, China; oldmario@126.com

**Keywords:** fucoidan, *Saccharina japonica*, anti-inflammation, mechanism

## Abstract

Fucoidan has been reported to have a variety of biological activities. However, different algae species, extraction methods, harvesting seasons, and growth regions lead to the structural variation of fucoidan, which would affect the bioactivities of fucoidan. To date, the anti-inflammatory properties and the underlying mechanism of fucoidan from brown alga *Saccharina japonica* (*S. japonica*) remain limited. The aims of the present study were to investigate the structure, the anti-inflammatory properties, and the potential molecular mechanisms of fucoidan isolated from *S. japonica* (SF6) against lipopolysaccharide (LPS)-activated RAW 264.7 macrophages. SF6 was characterized using high performance liquid gel permeation chromatography (HPGPC), Fourier transform infrared spectroscopy (FTIR), and nuclear magnetic resonance spectroscopy (NMR), and observed to be rich in fucose, galactose, and sulfate. Additionally, results showed that SF6 remarkably inhibited LPS-induced production of various inflammatory mediators and pro-inflammation cytokines, including nitric oxide (NO), NO synthase (iNOS), cyclooxygenase-2 (COX-2), tumor necrosis factor-α (TNF-α), interleukin-β (IL-β), and interleukin-6 (IL-6). A mechanism study showed that SF6 could effectively inhibit inflammatory responses through blocking LPS-induced inflammation pathways, including nuclear factor-κB (NF-κB), mitogen-activated protein kinase (MAPK), and Janus kinase (JAK)-2 and signal transducer and activator of transcription (STAT)-1/3 pathways. These results suggested that SF6 has the potential to be developed as an anti-inflammatory agent applied in functional food.

## 1. Introduction

*Saccharina japonica* is the most productive edible brown algae in China and has been used as a traditional medicine for thousands of years in China. Nowadays, *S. japonica* is a well-known source of bioactive compounds, including mannitol, alginates, fucoidan, and laminarian, in which fucoidan as the major active component has attracted widespread attention in recent years [1,2]. Fucoidan is a kind of fucose-containing sulfated polysaccharide, found in the fiber pile cell walls and intercellular spaces of brown seaweeds and echinoderm [3]. It has a number of attractive biological activities, such as antiviral [4], anticancer [5,6], anti-inflammatory [7,8], antioxidant [9], hypolipidemic [10,11], and immunostimulatory effects [12].

Inflammation, a host defense response to tissue injuries, infection, stress, and other stimuli, plays a critical role in homeostasis and fine regulation. However, excessive and continuous inflammatory responses turn out to be very harmful to the host, leading to tissue damage from diverse diseases, including obesity, arthritis, cancer, autoimmune diseases, etc. [3,13,14]. The various biological activities of fucoidan, especially in the prevention of inflammation-related diseases and related molecular mechanisms, have attracted great interest. Fucoidan could exert anti-inflammatory effects by inhibition of LPS-induced expression of inflammatory mediators and pro-inflammatory cytokines and down-regulation the protein expression levels of iNOS and COX-2 in macrophage cells. For example, Kang et al., found that the fucoidan extracted from *Ecklonia cava* significantly inhibited NO production and prostaglandin-E2 (PGE2) production, and suppressed inducible iNOS and COX-2 expression in LPS-stimulated RAW 264.7 cells. Additionally, the fucoidan from *Sargassum horneri* showed high inhibition of NO production in LPS-stimulated RAW 264.7 cells and down-regulated the protein expression levels of iNOS and COX-2 and the production of inflammatory cytokines, including TNF-α and IL-1β [15].

The underlying anti-inflammatory mechanism may be correlated with the suppression of the activation of NF-κB and the MAPKs signal pathways [7,15,16,17]. It was found that fucoidan purified from *Fucus vesiculosus* exhibited anti-inflammatory properties by suppression of NF-κB activation and down-regulation of MAPKs and Akt pathways in microglial cells [18]. In addition, fucoidan isolated from the brown seaweed *Padina commersonii* inhibited LPS-induced inflammatory responses via blocking TLR/MyD88/NF-κB signal transduction [7]. A recent study indicated a fucoidan (LJSF4) purified from *S. japonica* was found to show a strong anti-inflammatory effect in LPS-induced RAW 264.7 macrophage cells and zebrafish. The mechanism was revealed to be associated with the down-regulated expression of signal pathways, including MAPK and NF-κB [19]. The structures of fucoidan vary, including the monosaccharide compositions, the position glycoside bonds, branched chains, sulfate radical content, substitution of sulfate groups position, degree of sulfation, and molecular weight, which are mainly influenced by the species, algae characteristics, geographical location, harvest season, extraction conditions, and other factors [20,21,22,23]. Considering the above-mentioned influence factors, every new fucoidan obtained could potentially be a new compound with unique structural characteristics and have promising bioactive properties.

In this study, four homogeneous fucoidan were isolated and purified from *S. japonica* in Fujian Province, southeast of China. Among them, SF6 was observed to have efficient anti-inflammatory effect during the in vitro bioactivity test. On this basis, the preliminary structure of SF6 was characterized by FTIR and 1D-NMR, and the anti-inflammatory effect was investigated in LPS-activated RAW 264.7 macrophage cells. Furthermore, the possible anti-inflammatory mechanism including NF-κB, MAPKs, and JAK2-STAT 1/3 signal pathways were comprehensively investigated.

## 2. Results

### 2.1. Yield and Physicochemical Properties of Polysaccharides Isolated from S. japonica 

In the present study, fucose containing sulfated polysaccharide was extracted from brown seaweed *S. japonica*; then the crude polysaccharide fraction with a yield of 8.34 ± 1.22% was further purified using a Q sepharose fast flow column. As shown in Figure 1A, seven fractions were obtained with the yields of 0.5%, 0.87%, 1.67%, 3.33%, 5.33%, 6.67%, and 1.5%, respectively. The purity and molecular weights of the seven fractions were detected by CAME (Figure 1B). Among them, four independent acidic polysaccharides, namely, SF3, SF4, SF5, and SF6, exhibited a single dot, suggesting they were in high purity. Then these four fractions were selected for further study. The chemical content, monosaccharide composition, and molecular weight of four fractions are shown in Table 1. SF6 was mainly composed of carbohydrates (58.17%) and sulfates (36.94%). The result of monosaccharide composition indicated that SF6 was mainly composed of mannose (17.87%), rhamnose (16.42%), galactose (41.54%), and fucose (24.15%). The average molecular weight of SF6 was 246.4 KDa. As could be seen in Table 1, the monosaccharide composition of the SF6 was simpler than the other three fractions, and complex monosaccharide composition enhanced the difficulties of elucidation for the structure of polysaccharide; therefore, structure analysis for SF6 is more applicable.

The anti-inflammatory activity of the four fractions were tested using macrophage RAW 264.7 cells (data not shown), among them, SF6 displayed great anti-inflammatory potential, subsequently, the structure and anti-inflammatory mechanism studies of SF6 were performed and described as follows.

### 2.2. Structural Feature of SF6

SF6 was further analyzed using FT-IR and NMR. The FT-IR spectra of SF6 showed typical polysaccharide absorption bands (Figure 2A). The bands of 3600–3200 cm^−1^ contributed to the deformation of O-H. The peak at approximately 2933 cm^−1^ was attributed to asymmetric and symmetric stretching vibration of the methylene groups. An absorption band at the 1733 cm^−1^ revealed the presence of acetyl groups [24,25]. The absorption peak at 1650 cm^−1^ is attributed to vibration modes of glycosaminoglycans [26]. The presence of uronic acids was evidenced by the characteristic signal band around 1433 cm^−1^ corresponding to C-O stretching vibration. The band at 1250 cm^−1^ assigned to sulfate was prominent [27]. The absorption at 850–818 cm^−1^ confirmed the presence of sulfates in SF6. The week shoulder at 818 cm^−1^ suggested that the polysaccharide was sulfated at C2 and/or C3 on fucose or C2 on galactose residues at equatorial position, and the absorption peak at 850 cm^−1^ suggested a sulfate substituent primarily at the axial C4 position in fucose, and the configuration could be α-anomeric [24,28]. Additionally, the peak at 1049 cm^−1^ corresponded to the C-O-C/C-OH stretching frequency.

The structure of SF6 was further investigated by NMR. The ^1^H NMR spectra (Figure 2B) of SF6 contained several intense signals in the α-anomeric (4.80–5.50 ppm) and high-field (1.00–2.50 ppm) regions. The signals at region of 1.00–1.30 ppm were assigned to C6 methyl protons of α-l-fucopyranose [29]; meanwhile, signals at 1.39 ppm and 1.95–2.15 ppm arose from -CH_3_ protons of N-acetyl and O-acetyl groups [26,30], respectively. The heavily overlapped proton signals at 3.20–4.80 ppm were assigned to H2-H6 of the sugar residues. In addition, the broadened signal peaks between 4.80–5.50 ppm usually represented the anomeric protons of α-l-fucopyranosyl units. In Figure 2B, there were two broadened signal peaks, one of which corresponded to a non-sulfated fucoidan between 4.8–5.1 ppm and the other one locating at approximately 5.3 ppm was the signal of the sulfated fucose residue.

The ^13^C NMR spectra (Figure 2C) of natural fucoidan SF6 contained several intense signals in the anomeric carbon region between 99.6–101.8 ppm and the high-field regions of 14–16 ppm corresponded to the C1 and C6 carbons of α-l-fucopyranosides, respectively. In addition, the signals at 60.2 and 61.1 ppm were assigned to unsubstituted C6 galactose residue (CH_2_OH-β-d-galactopyranose) [5], while signal at 66.4 ppm was attributed to the glycosylated C6 of the β-d-galacylpyranose residue (CH_2_OR-β-d-galactopyranose), and the signals at 103.3–104.2 ppm were attributed to the C1 of β-d-galactopyranose. Several carbonyl carbon signals appeared at 175.0–177.0 ppm and corresponding methyl carbon signals at 21.1 and 25.5 ppm were assigned to O-acetyl groups and N-acetyl groups, respectively [26,31].

### 2.3. Effect of SF6 on Macrophage Viability

The effect of the SF6 on the viability of RAW 264.7 is shown in Figure 3. The survival rates of macrophage cells exposed to SF6 at the concentration of 50–200 μg/mL were all above 90%, suggesting RAW 264.7 cells were not affected by SF6 at each concentration. The result indicated that SF6 had no cytotoxicity on RAW 264.7 cell at concentration up to 200 μg/mL.

### 2.4. Effect of SF Fractions on NO Release in LPS-Stimulated RAW 264.7 Macrophage 

The intracellular NO release in RAW 264.7 macrophages was measured with a commercial kit. The effect of SF6 on the release of NO in the macrophage cells is shown in Figure 4. The result showed that SF6 exhibited a considerable NO inhibition effect. The effect of SF6 on the release of NO at the concentration of 200 μg/mL was almost the same as that of dexamethasone (DEM). Therefore, SF6 was selected for the subsequent anti-inflammatory experiment.

### 2.5. Effect of SF6 on iNOS and COX-2 mRNA and Protein Expression in LPS-Induced RAW 264.7 Cells

The mRNA and protein expressions of iNOS and COX-2 were determined to figure out the relationship between the inhibitory effect of SF6 on NO production and the expression levels of iNOS and COX-2. The results in Figure 5 showed SF6 attenuated the elevated levels of mRNA express of iNOS and COX-2 compared with LPS-stimulated RAW 264.7 cells. In addition, LPS treatment resulted in a significant up-regulation of the protein express levels of iNOS and COX-2, whereas SF6 remarkably down-regulated the iNOS and COX-2 protein levels. The results suggested that the inhibitory effect of SF6 on NO and PGE2 production was connected with the reduced levels of iNOS and COX-2 expression.

### 2.6. Inhibitory Effect of SF6 on Pro-Inflammatory Cytokine Secretion in LPS-Stimulated RAW 264.7 Cells

The secretion of the pro-inflammatory cytokines, including TNF-α, IL-6, and IL-1β, in macrophage cells was measured to evaluate the anti-inflammatory potential of SF6. As shown in Figure 6, LPS exposure resulted in a remarkable increase in the productions of TNF-α, IL-6, and IL-1β compared with the untreated control cells, while SF6 treatment significantly inhibited the secretion of those cytokines. The results suggested that SF6 exerted anti-inflammatory activity involved in decreasing the production of pro-inflammatory cytokines.

### 2.7. Effect of SF6 on NF-κB Activation in LPS-Induced RAW 264.7 Cells

The effect of SF6 on the NF-κB signaling pathway was evaluated to assess whether the suppressing effect of SF6 on the secretion of pro-inflammatory cytokines was mediated through the inhibition of NF-κB pathway in LPS-induced macrophages. As shown in Figure 7, the increased phosphorylation levels of IKK-α and IκB-α were observed in LPS-activated macrophages. In addition, the phosphorylation levels of NF-κB (p50 and p65) and the nucleus translocation levels of p50 and p65 also remarkably increased by LPS stimulation as expected. Notably, treatment of SF6 remarkably inhibited the phosphorylation and proteolytic degradation of cytoplasmic IKK-α, IκB-α, p50, and p65 in macrophages RAW 264.7. Concurrently, nucleus translocation levels of NF-κB (p50 and p65) were also down-regulated by SF6 in LPS-activated RAW 264.7 cells. Together, these results indicated that the inhibitory effect of SF6 on LPS-induced inflammatory process was partly attributed to the suppression of NF-κB activation in RAW 264.7 cells.

### 2.8. Effect of SF6 on the Phosphorylation of MAPKs in LPS-Stimulated RAW 264.7 Macrophages

To determine the effect of SF6 on the MAPKs signaling pathway, we further evaluated the activation levels of the MAPKs pathway’s components in LPS-stimulated RAW 264.7 cells via Western blot. As shown in the results (Figure 8), LPS-induced high phosphorylation levels of ERK 1/2, JNK, and p38 MAPKs were remarkably suppressed by SF6 treatment; meanwhile, the total levels of ERK1/2, JNK, and p38 stayed the same in all groups. These findings revealed that SF6 might mediate the anti-inflammatory effect by suppressing the MAPKs inflammatory signaling pathway. 

### 2.9. Effect of SF6 on JAK2-STAT1/3 Pathway Activation in LPS-Induced RAW 264.7 Cells

Furthermore, the molecular mechanism of SF6 on JAK2/STATs signaling pathway was evaluated. Specifically, the phosphorylation levels of STAT1/3 and JAK2, a key kinase of STAT1/3, were investigated. As shown in Figure 9, LPS stimulation significantly promoted phosphorylation levels of JAK2 and STAT1/3 compared to the control group. However, the up-regulated levels of JAK2 and STAT1/3 were significantly down-regulated by SF6 treatment, suggesting that SF6 might target JAK2 and further block the activation of phosphorylated STAT1/3 to exert anti-inflammatory effect.

## 3. Discussion

Studies have found that inflammation, a defensive host reaction responding to pathogenic stimuli, plays a critical role in the pathogenesis of inflammation related diseases [7]. Our study demonstrated that SF6, a bioactive high sulfate content fucoidan isolated and purified from *S. japonica*, could inhibit LPS-activated inflammation via down-regulation of various inflammatory mediators. Recent studies have shown that fucoidan displays a variety of activities, including anticancer, anti-inflammatory, hypolipidemic, and immunostimulatory effects [9,13]. For example, *Undaria pinnatifida* derived fucoidan attenuated LPS-stimulated inflammation in RAW 264.7 macrophage cells by inhibiting the phosphorylation of MAPK (ERK, p38, and JNK) [17]. Li et al. found that administration of fucoidan isolated from *Laminaria japonica* could regulate the inflammation response via HMGB1 and NF-κB inactivation in the ischemia–reperfusion-induced myocardial damage model [32]. In addition, fucoidan from sea cucumber *Pearsonothuria graeffei* exerts powerful effects in terms of reducing obesity and improving lipid profile by regulating gut microbiota [33]. A recent study indicated a fucoidan (LJSF4) purified from *S. japonica* displayed a strong anti-inflammatory effect in vitro and in vivo by down-regulation of signal pathways, including MAPK and NF-κB [19]. It is believed that every newly obtained fucoidan could potentially be a new compound with unique structural characteristics and promising bioactive properties; however, the molecular mechanism by which fucoidan from *S. japonica* inhibits the inflammatory process remains uncomprehensive. In the present study, SF6 from *S. japonica* was found to effectively inhibit inflammatory responses through blocking LPS-induced inflammation pathways, including NF-κB, MAPK, and JAK2-STAT1/3 pathways. These results suggested that SF6 exerted effective pharmacological activities for inflammation inhibition.

In our study, four fractions of fucoidan were separated and purified from *S. japonica* cultivated in southeast part of China; among them, SF6 is a fucoidan rich in fucose (24.15%), galactose (41.54%), and sulfate (36.94%). A similar monosaccharide composition was observed in fucoidans of other brown seaweeds. Fucoidans extracted from the *Padina commersonii* predominantly contained fucose and galactose, and also contained a sulfate group plus small amounts of mannose, rhamnose, and xylose [7]. Similarly, fucoidan fractions isolated from *Saccharina sculpera* mainly consisted of fucose and galactose, with small amounts of mannose, glucose, rhamnose, xylose, and glucuronic acid [11]. In addition, Dai et al. found that fucoidan from *Hizikia fusiforme* consisted of fucose (37.56%), galactose (38.43%), mannose (22.55%), rhamnose (1.05%), and arabinose (0.40%) [34]. Sulfate and fucose are two critical factors with which to determine the bioactivity of fucoidan. Recent studies demonstrated that fucoidan with higher contents of sulfates and fucose usually has higher bioactivities than those with lower content [35,36]. Alboofetileh et al. found that fucoidan isolated from *Nizamuddinia zanardinii* with greater sulfate content had higher anticancer and immunostimulatory activities [37]. The study of molecular characteristics and anti-inflammatory activity of three fucoidans from *Ecklonia cava* suggested that the level of NO released from macrophages was proportionally related to the sulfate and fucose content of the fucoidans [38]. SF6 has the highest sulfate content in all the four obtained fucoidan fractions. Although the content of fucose of SF6 is not the highest one, SF6 still had better anti-inflammation than the other three fractions. Furthermore, FTIR and NMR were performed to preliminarily characterize the structure of SF6, indicating it presented the specific features of natural fucoidan.

In general, abundance of pro-inflammatory cytokines including TNF-α, IL-6, and IL-1β would be over-expressed during the process of inflammation; therefore, the levels of these pro-inflammatory cytokines are usually regarded as indicators of the degree of inflammation [39]. Thus, the suppression of excessive production of the inflammatory mediators is considered to be an effective way to prevent the occurrence and development of inflammatory process. Exhilaratingly, our study showed that SF6 obviously inhibited the LPS-induced production of TNF-α, IL-6, and IL-1β in RAW 264.7 cells.

Nitric oxide, an important pro-inflammatory mediator, is mainly synthesized by iNOS enzyme which would excessive express in reaction to various immune and inflammatory stimuli. The excessively produced NO could result in the generation of various pro-inflammatory mediators and cytokines, which would further promote the development of inflammation and finally relate to tissue damage and detrimental inflammatory disorders [40]. The high level of iNOS is often accompanied with over-expressed COX-2 in the inflammatory process, and COX-2 is the key enzyme required for the conversion of arachidonic acid into PGE2 which is a characteristic marker of inflammatory damage [7]. In the current study, the excessive level of NO in the LPS-stimulated macrophages was considerably down-regulated by the treatment of SF6 at various concentrations, which could be attributed to the inhibition of protein expression of iNOS and COX-2. Therefore, SF6 could exert its anti-inflammatory effect at least in part through suppression of excessive production of inflammation mediators and pro-inflammation cytokines including TNF-α, IL-6, IL-1β, NO, iNOS, and COX-2.

A number of studies indicated that LPS binding to the surface of macrophages activates a series of complex intracellular inflammatory cascade signaling pathways, including the NF-κB signal pathway [7]. It plays a critical role in inflammatory diseases because the activation of NF-κB can induce the over-expression of inflammatory genes and further promote the production of inflammatory mediators and pro-inflammation cytokines such as NO, iNOS, COX-2, TNF-α, IL-6, and IL-1β [41,42,43]. Generally, NF-κB, localizing in the cytoplasm, presents as a complex with IκB in an inactive form. Upon inflammatory activation, the upstream kinase IKK mediates the phosphorylation and degradation of IκB, which leads to the release and translocation of NF-κB into the nucleus and results in the inflammatory genes transcription [7,44,45]. In our study, the effect of SF6 on NF-κB signal pathway was examined; the result indicated the phosphorylation level of NF-κB in the nucleus was remarkably increased by LPS stimulation, but nevertheless, pre-treatment with SF6 effectively inhibited the translocation of NF-κB into the nucleus. Therefore, SF6 could effectively block the activation of NF-κB pathway.

The MAPKs, as the upstream inflammatory signal pathway, are also involved in the process of inflammation. Both in vitro and in vivo researches have shown that activation of MAPKs (ERK, JNK, and p38) can mediate the gene transcriptions in the inflammatory responses to LPS [16,42,45]. As reported that the activities of p38 and JNK MAPKs are significantly increased in macrophages stimulated by LPS through enhancing their phosphorylation levels, and then accelerate inflammatory process [42,46]. In addition, ERK1/2 is identified to be directly related to NF-κB activation [44]. The present study displayed that the increased phosphorylation levels of ERK1/2, JNK, and p38 MAPKs were effectively suppressed by SF6 treatment.

In addition to NF-κB and MAPKs signal pathways, previous studies indicated that JAK-STAT pathway is also involved in LPS-induced over-expression of inflammation mediators and pro-inflammation cytokines, especially iNOS, IL-6, and IL-1β, in macrophage cells. Upon inflammatory activation, membrane bound JAK receptor proteins enhance the phosphorylation of the major substrate, the family of STATs, including STAT1 and STAT3. Phosphorylated STAT1/3 can translocate to the nuclear and then result in the expression of pro-inflammatory genes [47,48,49]. In this study, SF6 treatment could significantly down-regulate JAK2 phosphorylation and nuclear translocation of STAT 1/3, indicating that SF6 could prevent the activation of JAK2 and then inhibit JAK2-STAT1/3 pathway.

## 4. Materials and Methods

### 4.1. Materials

Fucoidan is purified from *S. japonica,* which was collected in September 2016 at Quangang, Fujian province, China. The plant material was authenticated by Prof. Meitian Xiao from Huaqiao University. A voucher specimen (number 20160906) was deposited at the Herbarium of Huaqiao University. LPS (L2880 055: B5), DMSO, dextran analytical standard (410, 270, 150, 50, 25, 5 kDa), d-(+)-glucosamine hydrochloride, d-(+)-galactosamine were purchased from Sigma-Aldrich (Sigma-Aldrich, St. Louis, MO, USA). Dulbecco’s modified Eagle’s medium (DMEM) containing l-glutamine (200 mg/L), fetal bovine serum (FBS), 3-(4, 5-dimethylthiazol-2-yl-)2.5-diphenyltetrazolium bromide (MTT) were purchased from Corning cellgro (Corning, New York, NY, USA). d-(+)-galactose, d-(+)-xylose, l-(+)-rhamnose hydrate, d-(+)-mannose were purchased from Dr. Ehrenstorfer GmbH (Dr. Ehrenstorfer GmbH, Augsburg, Germany), d-(+)-glucose was purchased from Aladdin (Aladdin, Shanghai, China), l-(−)-fucose was purchased from NIFDC (Beijing, China). Other chemicals were of analytical grade. Primary antibodies were purchased from Cell Signaling Technology (Beverly, MA, USA). The second antibody was bought from Abcam (Cambridge, UK). Chemiluminescent HRP was purchased from Pierce Scientific (Rockford, IL, USA). Nitric oxide assay kit and the nuclear and cytoplasmic protein extraction kit were purchased from Beyotime Biotech (Guangzhou, China). TNF-α, IL-1β, and IL-6 ELISA kits were from ExCell Bio company (Shanghai, China).

### 4.2. Isolation and Purification of the Polysaccharide from Saccharina japonica

The *S. japonica* samples were washed, dried, and ground to powder, 50.0 g of the dried powder was suspended in 95% ethanol for 4 h under boiling point with continuous agitation to remove impurities such as mannitol, ester, and pigment. Then the sample was washed with ethanol for three times. After filtration, the remains were dried in a drying oven. The dried algae were dipped into distilled water and kept at 92 °C for 4 h. After that, the supernatant was separated from algae residues through filtration and gradually treated with 4% CaCl_2_ facilitating the precipitation of any alginic acid impurities. The supernatant was filtered and concentrated to one-eighth of its original volume by rotary evaporator. Finally, four volumes of 95% ethanol were added to the concentrated solution and kept at 4 °C overnight. Polysaccharides were recovered by centrifugation at 10,000× *g* and washed with ethyl alcohol for three times. The obtained crude polysaccharide was re-dissolved in distilled water and added with ethanol to a concentration of 30% to remove alginate, and subsequently to 70% to obtain a primary purified fucoidan (SF). SF was fractionated by a Q sepharose Fast flow column with distilled water and 0.1–2.0 M NaCl stepwise gradient solution, giving the fraction of SF3, SF4, SF5, SF6, respectively. The polysaccharide content in each tube was determined by the phenol-sulfuric assay method and pooled based on the total carbohydrate content. The salt in the eluate was removed by using a 3.5 kDa molecular weight cut-off dialysis tubing (Spectra/Por USA) and then lyophilized.

### 4.3. Homogeneity and Molecular Weight of the Purified Fucoidan

The purity of SF fractions was determined by cellulose acetate membrane electrophoresis (CAME). Briefly, cellulose acetate membranes (2 × 8 cm) were immersed in 0.1 mol/L HCl solution for 30 min. After the excess buffer was removed, the electrophoresis was carried out under 2.5 V/cm, 0.4–0.6 mA/cm for 200 min. The membrane was stained with 0.2% Alcian blue solution containing 10% ethanol, 0.1% glacial acetic acid and 0.03 M MgCl_2_ [50].

The molecular weights (Mw) of the purified fucoidans were determined by high performance liquid gel permeation chromatography (HPGPC) with a Shodex KS-804 column (8 mm × 300 mm, 7 μm) at 80 °C. Additionally, the sample was eluted with highly purified water at a flow rate of 0.5 mL/min. The molecular weight of polysaccharides was calculated by a standard curve obtained using dextran of 410, 270, 150, 50, 25, 5 kDa as reference standards.

### 4.4. Composition Analysis

Accordingly, phenol-sulfuric method for measuring the polysaccharide content [51] and barium sulfate precipitation method for sulfate content were used during the analysis [52]. The amount of proteins was determined according to the folin-phenol assay using the bovine serum albumin (BSA) standard curve [53]. The composition of monosaccharide was determined by HPLC chromatography after converting them into 1-phenyl-3-methyl-5-pyrazolone (PMP), derivatives [34]. Briefly, polysaccharide was hydrolyzed in a sealed glass tube with 2M trifluoroacetic acid (TFA) at 110 °C for 2 h; after that, TFA was removed under reduced pressure using a rotary evaporator with methanol and the hydrolysate was evaporated to dryness; 1 mL of distilled water was added to the hydrolysate, 100 μL of hydrolysate was taken into the glass tube and added with 100 μL of 0.5 M PMP in methanol and 0.3 M NaOH; then the solution was incubated at 70 °C for 1 h; after cooling down, 100 μL of 0.3 M HCl was added to adjust the pH to neutrality. Finally, excess chloroform was added; the sample was mixed by shaking; the separation of the two phases was done by allowing the mixture to settle and the organic phase was discarded. Monosaccharide compositions’ identification was done by comparison with reference sugars. Calculation of the molar ratio of the monosaccharides was carried out on the basis of the peak area of the monosaccharides. HPLC chromatograph was performed on an LC-16 (Essentia, Japan) equipped with an Agela Venusil XBP-C18 column (5 μm, 250 × 4.6 mm) (Agela China); 0.1 M phosphate buffer (pH 6.7) and acetonitrile with a ratio of 80:20 at a flow rate of 0.5 mL/min were used as elution solvents and detection wavelength was set at 250 nm; the sample injection volume was 20 μL.

### 4.5. Fourier Transform Infrared Characterization

The SF6 was analyzed with Fourier transform infrared (FT-IR) spectrometer (Thermo Scientific Nicolet iS 50, MA, USA) by KBr method. Scans were collected with an average of 32 scans and a spectral range and resolution of 400–4000 cm^−1^ and 4 cm^−1^, respectively.

### 4.6. NMR Spectroscopy

NMR spectra were recorded with a Bruker Avance III 500 spectrometer at 293 K. Samples were deuterium-exchanged by lyophilization with D_2_O, and dissolved in D_2_O with deuterated acetone as the internal standard. The concentration of the sample was 30 mg of polysaccharide/500 μL D_2_O.

### 4.7. Biological Assays

#### 4.7.1. Cell Culture

Mouse mononuclear macrophages leukemia RAW 264.7 cell line was obtained from Chinese Academy of Medical Science and Peking Union Medical College (Beijing, China) and maintained in Dulbecco’s modified Eagle’s medium (DMEM), supplemented with 10% fetal bovine serum, 100 U/mL penicillin, and 100 μg/mL streptomycin in a humidified atmosphere of 5% CO_2_ at 37 °C.

#### 4.7.2. Measurement of Cell Viability of RAW 264.7 Macrophages

The RAW 264.7 cells of logarithmic phase were collected and seeded at a density of 5 × 104 cells/well in 96-well plates and the cell viability was measured using the MTT assay. After pre-incubation of RAW 264.7 cells with different concentrations of sample (50, 100, 150, 200 μg/mL) at 37 °C for 24 h, the cells were incubated with 10 μL of 5 mg/mL MTT in culture medium at 37 °C for another 4 h. Then, 150 μL of DMSO was added and the absorbance was measured at 570 nm. The viability of RAW 264.7 macrophage in each well was presented as percentage of control cells.

#### 4.7.3. Measurement of NO Production

After pre-incubation of RAW 264.7 cells (5 × 10^4^ cells/well in 24-well plates) with LPS (1 μg/mL) and samples at 37 °C for 24 h, then the culture supernatant was mixed with the same volume of Griess reagent and absorbance at 550 nm was measured for analytic purpose of NO release. As an index of NO, the accumulation amount of nitrite was determined from a standard curve with sodium nitrite.

#### 4.7.4. Measurement of Pro-Inflammatory Cytokines Production

The inhibitory effect of fucoidan samples on the pro-inflammatory cytokines (TNF-α, IL-1β, and IL-6) production from LPS-induced RAW 264.7 macrophages was determined by using mouse ELISA kits according to the manufacture’s protocol.

#### 4.7.5. Preparation of Cell Lysates

The macrophage cells of logarithmic phase were collected and seeded at a density of 5 × 10^4^ cells/well in 6-well plates. When the density of cells reached 90%, they were pre-treated with filtered culture medium with different concentrations of SF3, SF4, SF5, SF6 (50, 100, 150, 200 μg/mL) or DEM (10 ng/mL) for 2 h, and then treated with LPS (1 μg/mL) for 24 h in a humidified atmosphere of 5% CO_2_ at 37 °C. Upon completion of the incubation studies the culture supernatant was collected for analysis of cytokine sections, and the cells were washed three times with ice-cold phosphate-buffered saline (PBS, pH 7.2) and then scraped from the plates into ice-cold 1% PMSF lysis buffer. The protein concentration was determined by the bicinchoninic acid (BCA) method. Aliquots were stored at −80 °C for the detection for the MAPK (ERK 1/2, JNK, p38, P-ERK 1/2, P-JNK, P-p38) protein expression level.

#### 4.7.6. Western Blot Analysis

After treated with fucoidan and LPS, RAW 264.7 cells were harvested and washed with ice-cold phosphate-buffered saline (PBS, pH 7.2). The total proteins and nuclear or cytoplasmic proteins were extracted from the cells using Cell Lysis Reagent (Sigma) and ice-cold lysis buffer. The protein content in the supernatant was determined by using the BCA protein assay kit. Equal amounts of proteins were loaded and fractionated by SDS-PAGE and then transferred into polyvinylidene difluoride (PVDF) membranes (Bio-Rad). The membranes were blocked in 5% (*w*/*v*) skimmed milk and then incubated with specific primary antibodies overnight at 4 °C followed by incubating with horseradish peroxidase-conjugated secondary antibody. Finally, the blots were probed using enhanced chemiluminescence (ECL) and auto radiographed.

### 4.8. Statistical Analysis

Data were expressed as means ± SDs and examined for their statistical significance of difference with ANOVA and *t*-test by using SPSS 16.0. Values of *p* < 0.05 and *p* < 0.01 were considered to be statistically significant.

## 5. Conclusions

In conclusion, the present study provided evidence that SF6 exerted an anti-inflammatory effect through suppression of production of pro-inflammatory cytokines such as TNF-α, IL-1β, and IL-6 and other mediators, including iNOS and COX-2. The molecular mechanism research indicated that the anti-inflammatory effect of SF6 was attributed to the modulation of NF-κB, MAPKs, and JAK2-STAT1/3 signaling pathways. Additionally, SF6 has the potential to be developed as broad spectrum anti-inflammation drugs or health products. Further studies regarding the structure and bioactivities of fucoidans are in progress to determine correlation between the molecular structures and biological activities.

## Figures and Tables

**Figure 1 marinedrugs-18-00328-f001:**
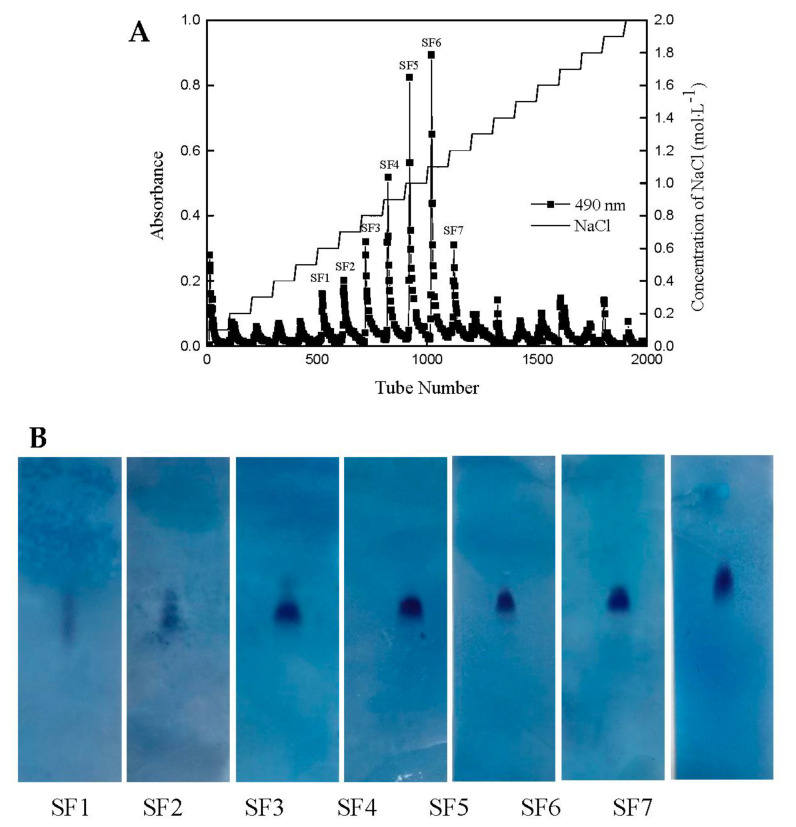
Purification of SF using Q sepharose fast flow chromatogram from *S. japonica*. (**A**) Elution curve of crude polysaccharide from *S. japonica*. (**B**) Cellulose acetate membrane electrophoresis of seven fractions of SF. SF was loaded to a Q sepharose fast flow column (600 mm × 30 mm) pre-equilibrated in distilled water. The column was eluted with an increasing gradient of NaCl (0.0–2.0 M). Seven fractions (SF1, SF2, SF3, SF4, SF5, SF6, and SF7) were identified by phenol-sulfuric assay.

**Figure 2 marinedrugs-18-00328-f002:**
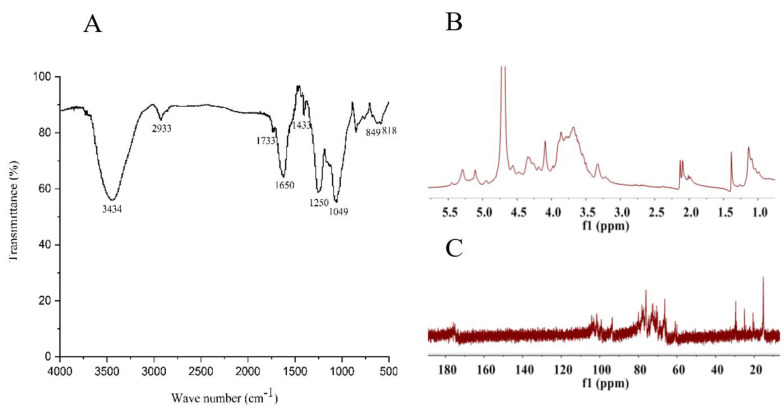
Characterization of the structural features of SF fractions using FT-IR and NMR. (**A**) FT-IR spectra of SF, SF3, SF4, SF5, and SF6. (**B**) ^1^H NMR spectra of SF6. (**C**) ^13^C NMR spectra of SF6.

**Figure 3 marinedrugs-18-00328-f003:**
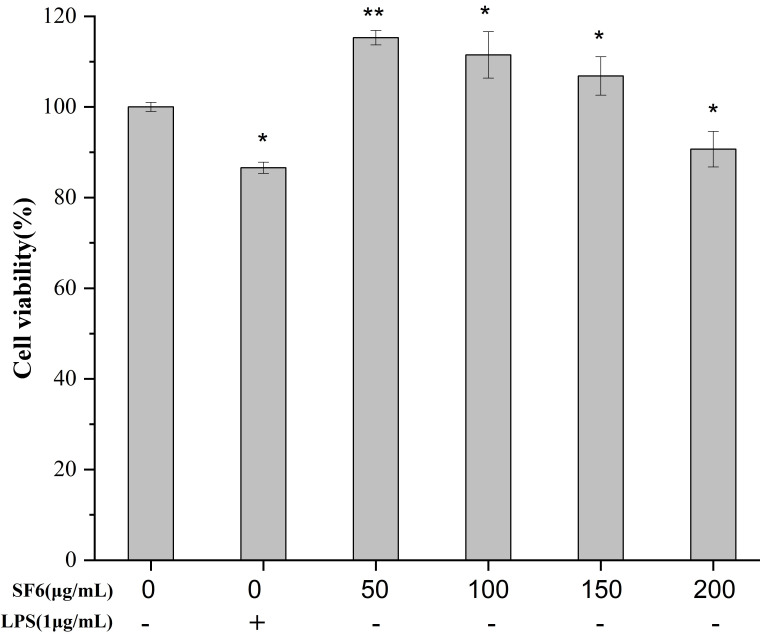
Effects of SF6 on the cell viability of RAW 264.7 macrophage. Cells (5 × 10^4^ cells/well in 96-well plates) were treated in different concentration of sample (50, 100, 150, 200 μg/mL) for 24 h; the cell viability was measured using the MTT assay. Values are presented as mean ± SD (*n* = 3) of three independent experiments. ** *p* < 0.01 vs. control.

**Figure 4 marinedrugs-18-00328-f004:**
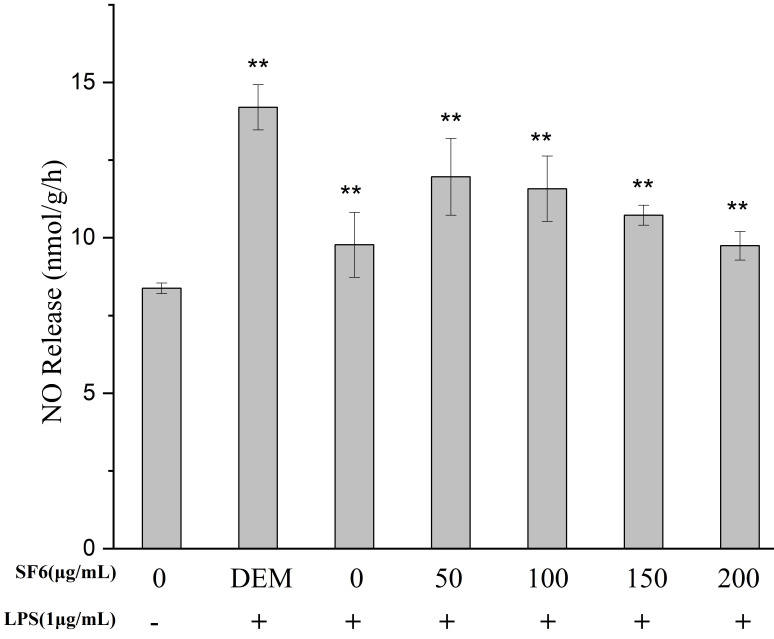
Effect of SF6 on NO release in LPS-stimulated RAW 264.7 macrophages. Cells (5 × 10^4^ cells/well in 24-well plates) were stimulated with LPS (1 μg/mL) for 24 h in the presence of DEM and SF6; the supernatant was collected for analysis of NO release. Values are presented as mean ± SD (*n* = 3) of three independent experiments. ** *p* < 0.01 vs. cells treated with LPS only.

**Figure 5 marinedrugs-18-00328-f005:**
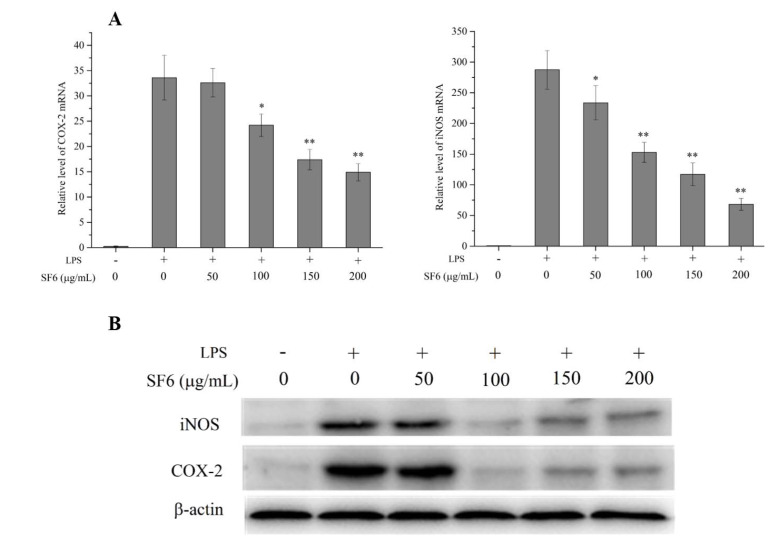
Effects of SF6 on the mRNA and protein expression of iNOS and COX-2 in LPS-stimulated RAW 264.7 cells. (**A**) The expression levels of iNOS and COX-2 mRNA were determined by RT-PCR. (**B**) The protein levels of iNOS and COX-2 were detected using western blot. Values are presented as mean ± SD (*n* = 3) of three independent experiments. * *p* < 0.05, ** *p* < 0.01 vs. cells treated with LPS only.

**Figure 6 marinedrugs-18-00328-f006:**
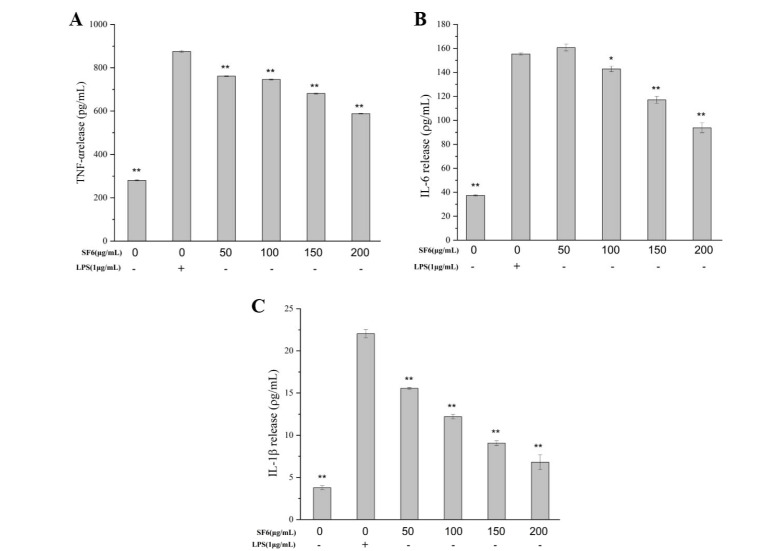
Effect of SF6 on the production of TNF-α, IL-1β, and IL-6 in LPS-stimulated RAW 264.7 macrophages. (**A**) The mRNA levels of TNF-α. (**B**) The mRNA levels of IL-6. (**C**) The mRNA levels of IL-1β. Each value indicated a mean ± SD from three independent experiments. * *p* < 0.05, ** *p* < 0.01 vs. cells treated with LPS only.

**Figure 7 marinedrugs-18-00328-f007:**
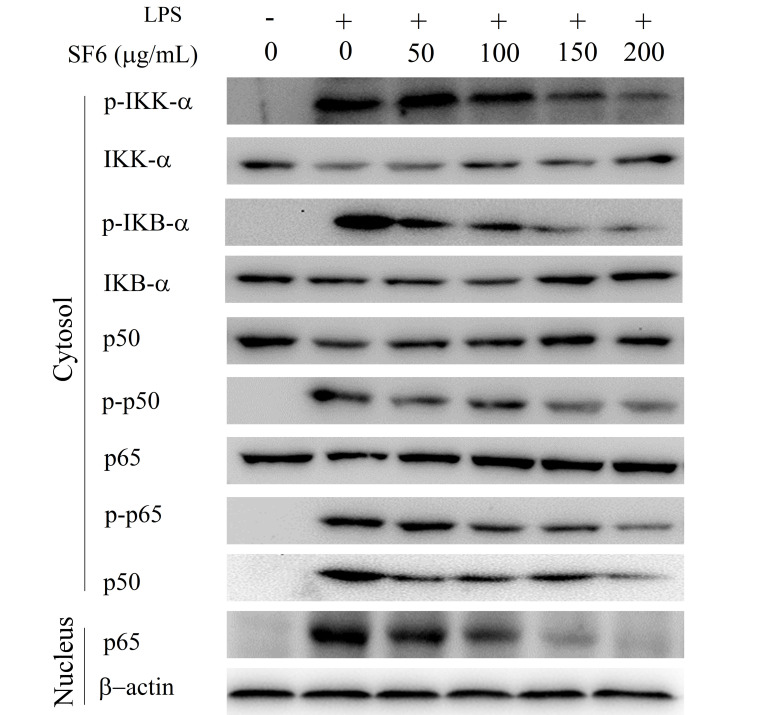
Effects of SF6 on the activation of the NF-κB signaling pathway in LPS-stimulated RAW 264.7 macrophages.

**Figure 8 marinedrugs-18-00328-f008:**
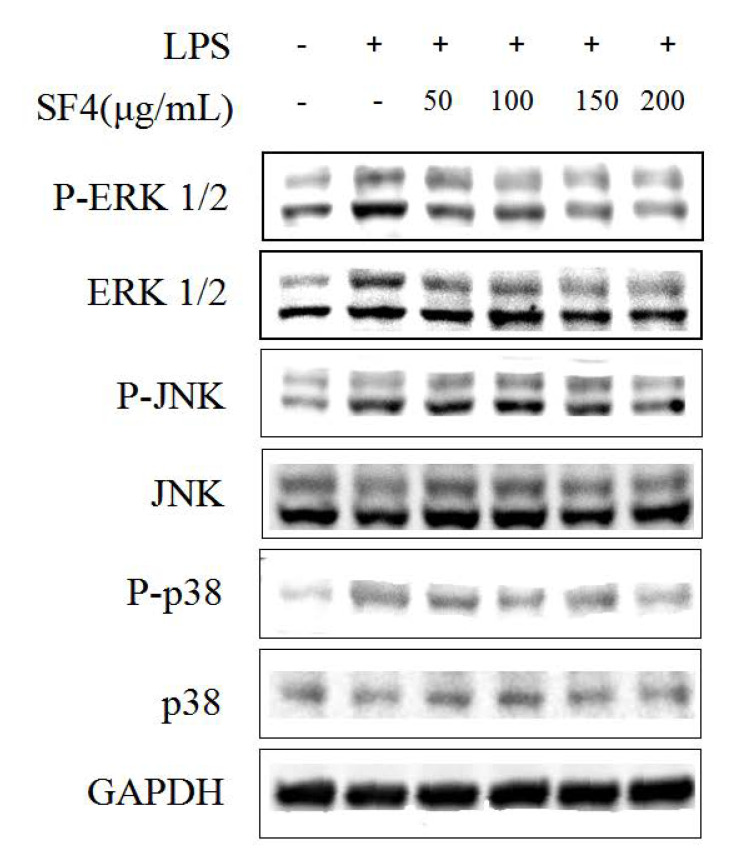
Effects of SF6 on the protein level sof ERK 1/2, JNK, and p38 MAPKs in LPS-stimulated RAW 264.7 macrophages.

**Figure 9 marinedrugs-18-00328-f009:**
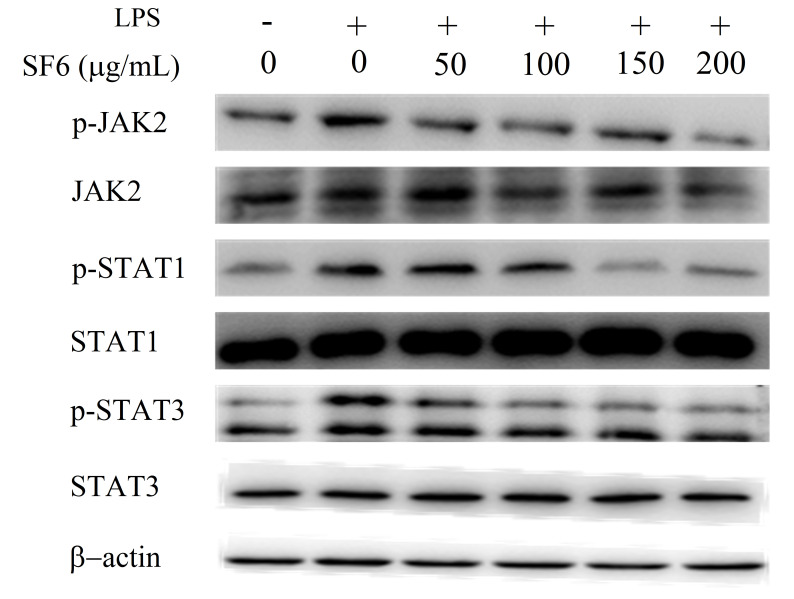
SF6 inhibited JAK2-STAT1/3 pathway activation in LPS-induced RAW 264.7 cells.

**Table 1 marinedrugs-18-00328-t001:** Chemical composition, monosaccharide composition and molecular weight of SF3, SF4, SF5, and SF6.

Sample	Total Sugar (%)	Sulfate (%)	Protein (%)	Monosaccharide Composition (%)					Molecular Weight (KDa)
				Mannose	Rhamnose	Galactose	Xylose	Fucose	
SF3	82.52 ± 2.42	12.05 ± 0.13	1.42 ± 0.28	24.22	2.34	32.81	1.56	39.06	222.2
SF4	79.63 ± 2.03	15.00 ± 0.07	1.17 ± 0.18	17.67	28.62	16.61	1.77	35.33	228.8
SF5	72.45 ± 1.98	23.79 ± 0.11	0.99 ± 0.06	11.02	24.80	22.83	1.97	39.37	231.5
SF6	58.17 ± 1.34	36.94 ± 0.08	1.02 ± 0.02	17.87	16.42	41.54	n.d.	24.15	246.4

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
