# Peer review of "Fucoidan Isolated from Saccharina japonica Inhibits LPS-Induced Inflammation in Macrophages via Blocking NF-κB, MAPK and JAK-STAT Pathways"

_marinedrugs, 2020, doi:10.3390/md18060328_

Round 1

Reviewer 1 Report

This is an interesting paper that evaluate the structure, the anti inflammatory properties as well as potential molecular mechanisms of fucoidan isolated from Saccharina japonica against LPS-activated RAW 264.7 macrophages

However, my specific concerns are as follows.

Major concerns:

  1. There are some significant grammatical and idiomatic errors in the text. Therefore, authors are urged to have this manuscript reviewed critically paying special attention to the English grammar.

Here are some examples:

-galatose=galactose (line 26)

-inflammatory-related diseases= inflammation-related diseases (line 49)

-was isolated= were isolated (line 66)

-survival rate ……………….were= survival rate was (line139)

- froctions=fractions (line 244)

- inflammatory gene= inflammatory genes (line 281 and 286)

-CaCl2= CaCl2 (line 330)

-etc.

  1. Authors along the manuscript report that the effect of SF6 was dose dependently. In order to confirm this, a statistical analyses comparing all the doses should be done. However, they have only compared each dose of SF6 with LPS group.
  2. Author report that the effect of SF6 on NO at the concentration of 200 μg/mL on RAW264.7 cells was even the same of DEM group. Please, check this sentence (Line 151)
  3. Line 165: “The results suggested that the inhibitory effect of SF6 on NO and PGE2 production was connected with the reduced levels of iNOS and COX-2 expression”. However, PGE2 production was not evaluated in this paper.
  4. In Figure 5, check bands for COX-2. There are not marked differences between them.
  5. Densitometry and statistical analysis for WB should be included in all Figures
  6. There is a recent paper: Ni L, Wang L, Fu X, Duan D, Jeon YJ, Xu J, Gao X. In vitro and in vivo anti-inflammatory activities of a fucose-rich fucoidan isolated from Saccharina japonica. Int J Biol Macromol. 2020 Apr 11;156:717-729. Authors should include this paper in their manuscript and discuss the novelty of their manuscript in relation with the above mentioned

Minor concerns:

  1. In Figure 3 legend: change # by *.
  2. Superscript and subscript should be reviewed in methods section
  3. Authors should indicate DEM meaning
  4. Authors should revise the abbreviations along the text.
  5. Line 406: “Different concentrations of SF3, SF4, SF5, SF6 (50, 100, 150, 200 μg/mL)….” Please, check because authors only used SF6
  6. Please, indicate in methods section the company for the antibodies.

Author Response

  • There are some significant grammatical and idiomatic errors in the text. Therefore, authors are urged to have this manuscript reviewed critically paying special attention to the English grammar.

Here are some examples: 

-galatose=galactose (line 26) 

-inflammatory-related diseases= inflammation-related diseases (line 49)

-was isolated= were isolated (line 66)

-survival rate ……………….were= survival rate was (line139)

- froctions=fractions (line 244)

- inflammatory gene= inflammatory genes (line 281 and 286)

-CaCl2= CaCl2 (line 330)

-etc.

Response: According to the reviewer’s comments, the grammatical and idiomatic errors were revised in the text, and all the changes were highlighted in red in the manuscript.

2.Authors along the manuscript report that the effect of SF6 was dose dependently. In order to confirm this, a statistical analyses comparing all the doses should be done. However, they have only compared each dose of SF6 with LPS group.

Response: According to the reviewer’s suggestion, we carried out the statistical analyses comparing all the doses of SF6, not all the results of SF6 on the production of pro-inflammatory cytokines showed the significant difference between the three doses of SF6, therefore, we removed the conclusion about dose dependence.

3.Author report that the effect of SF6 on NO at the concentration of 200 μg/mL on RAW264.7 cells was even the same of DEM group. Please, check this sentence (Line 151)

Response: According to the reviewer’s comments, the sentence of “The effect of SF6 on NO at the concentration of 200 μg/mL on RAW264.7 cells was even the same of DEM group.” was revised to “The effect of SF6 on the release of NO at the concentration of 200 μg/mL was almost the same as that of dexamethasone (DEM).”.

  1. Line 165: “The results suggested that the inhibitory effect of SF6 on NO and PGE2 production was connected with the reduced levels of iNOS and COX-2 expression”. However, PGE2 production was not evaluated in this paper.

Response: According to the reviewer’s comments, the PGE2 was removed in the main text.

  1. In Figure 5, check bands for COX-2. There are not marked differences between them.

Response: We had the western blot of COX-2 redone, and the result was shown in figure 5, there are significant differences between 100, 150, and 200 μg/mL SF6 groups and LPS group.

  1. Densitometry and statistical analysis for WB should be included in all Figures

Response: The densitometry and statistical analysis for all the WB experiments were included in supporting information, because the figures would be too big to include in the main text.

  1. There is a recent paper: Ni L, Wang L, Fu X, Duan D, Jeon YJ, Xu J, Gao X. In vitro and in vivo anti-inflammatory activities of a fucose-rich fucoidan isolated from Saccharina japonica. Int J Biol Macromol. 2020 Apr 11;156:717-729. Authors should include this paper in their manuscript and discuss the novelty of their manuscript in relation with the above mentioned

Response: According to the reviewer’s comments, we have added the paper (Int J Biol Macromol. 2020 Apr 11;156:717-729.) in our manuscript. It has been reported that the structures of fucoidan vary, which are mainly influenced by the species, algae characteristics, geographical location, harvest season, extraction conditions, and other factors. Considering these influence factors, every new obtained fucoidan could potentially be a new compound with unique structural characteristics and have promising bioactive properties. We think this is the novelty of our work.

Minor concerns:

  1. In Figure 3 legend: change # by *.

Response: The # was changed to *.

  1. Superscript and subscript should be reviewed in methods section

Response: According to the reviewer’s comments, we had reviewed all the superscript and subscript in the methods section and fixed the errors.

  1. Authors should indicate DEM meaning

Response: DEM is the abbreviation of dexamethasone, we indicated the full name in the first time it is referred in the text.

  1. Authors should revise the abbreviations along the text.

Response: We have revised all the abbreviations along the text, please see the highlights in red in the text.

  1. Line 406: “Different concentrations of SF3, SF4, SF5, SF6 (50, 100, 150, 200 μg/mL)….” Please, check because authors only used SF6

Response: SF3, SF4, and SF5 were removed from the sentence.

  1. Please, indicate in methods section the company for the antibodies.

Response: According to the reviewer’s comments, the companies of the primary and second antibodies were already added in the methods section. Please see “4.1 Materials.

Reviewer 2 Report

Authors studied the structure, the anti-inflammatory properties as well as potential molecular mechanisms of fucoidan isolated from Saccharina japonica. Although this is an interesting work, this subject has been addressed in many publications. So the novelty and significance of the research and the extent to which it adds to existing knowledge need to be highlighted. Other issues preclude publication.

All species should be written in Italic. In addition, the first time they are referred in the text it should be indicated by the full name and the corresponding authority; after that, only abbreviated forms should be used.

Table 1: Do total sugar, sulphate and protein values correspond to mean ± SD? How many determinations?

How was the test used for statistical analysis chosen? Information about normality of distribution and homogeneity of variance of the data should be provided.

I would prefer to read some more personal remarks and conclusions on the development and commercialization, as well as on the challenges that need to be faced to further exploit fucoidan as nutraceutical/pharmaceutical, emphasizing the window of opportunities/challenges arising from what they learnt.

I find issues regarding English that should be addressed with the help of a native speaker or English language editing service

Author Response

  1. Authors studied the structure, the anti-inflammatory properties as well as potential molecular mechanisms of fucoidan isolated from Saccharina japonica. Although this is an interesting work, this subject has been addressed in many publications. So the novelty and significance of the research and the extent to which it adds to existing knowledge need to be highlighted. Other issues preclude publication.

Response: Thanks for the comments. It has been reported that the structures of fucoidan vary, which are mainly influenced by the species, algae characteristics, geographical location, harvest season, extraction conditions, and other factors. Considering these influence factors, every new obtained fucoidan could potentially be a new compound with unique structural characteristics and have promising bioactive properties. We think this is the novelty of our work.

  1. All species should be written in Italic. In addition, the first time they are referred in the text it should be indicated by the full name and the corresponding authority; after that, only abbreviated forms should be used.

Response: All species have been revised to be written in Italic and we have revised all the abbreviations along the text.

  1. Table 1: Do total sugar, sulphate and protein values correspond to mean ± SD? How many determinations?

Response: Yes, all of the analyses were performed in triplicate.

  1. How was the test used for statistical analysis chosen? Information about normality of distribution and homogeneity of variance of the data should be provided.

Response: The data examined for their statistical significance of difference with ANOVA and t-test by using SPSS 16.0 and data were expressed as means±SD.

  1. I would prefer to read some more personal remarks and conclusions on the development and commercialization, as well as on the challenges that need to be faced to further exploit fucoidan as nutraceutical/pharmaceutical, emphasizing the window of opportunities/challenges arising from what they learnt.

Response: Fucoidan has attracted great interests as a potential candidate for the development of nutraceutical or pharmaceutical. However, it still has a long way to go through. Till now, the researches mainly focused on the in vitro activities of fucoidan, the in vivo assessments on the activities of fucoidan are far from enough. Furthermore, most of pharmacological activities were tested using crude extracts or partially purified product, the reasons might be due to the complex nature of crude extract matrix, time consuming and expensive materials applied in chromatographic techniques. In addition, there are many other challenges including the structure’s heterogeneity, co-extracted contaminants, quality faced fucoidan’s development.

  1. I find issues regarding English that should be addressed with the help of a native speaker or English language editing service.

Response: We have asked a native speaker to go through the main text and all the changes were highlighted in red in the text.

Round 2

Reviewer 1 Report

Authors have reviewed the manuscript following reviewer´s suggestions. Only, please in line 65, change resent by recent.

Author Response

Thanks for the  reviewer's hard work and patience. We have revised resent to recent in line 65.